# Refining Low-Resource Unsupervised Translation by Language Disentanglement of Multilingual Model

**Xuan-Phi Nguyen**[1,3]**, Shafiq Joty**[1,2]**, Wu Kui**[3]**, Ai Ti Aw**[3]
[1]Nanyang Technological University
[2]Salesforce Research
[3]Institute for Infocomm Research (I2R), A*STAR
Singapore
{nguyenxu002@e.ntu,srjoty@ntu}.edu.sg
{wuk,aaiti}@i2r.a-star.edu.sg

## Abstract

Numerous recent work on unsupervised machine translation (UMT) implies that competent unsupervised translations of low-resource and unrelated languages, such as Nepali or Sinhala, are only possible if the model is trained in a massive multilingual environment, where these low-resource languages are mixed with high-resource counterparts. Nonetheless, while the high-resource languages greatly help kick-start the target low-resource translation tasks, the language discrepancy between them may hinder their further improvement. In this work, we propose a simple refinement procedure to separate languages from a pre-trained multilingual UMT model for it to focus on only the target low-resource task. Our method achieves the state of the art in the fully unsupervised translation tasks of English to Nepali, Sinhala, Gujarati, Latvian, Estonian and Kazakh, with BLEU score gains of 3.5, 3.5, 3.3, 4.1, 4.2, and 3.3, respectively. Our codebase is available at github.com/nxphi47/refine_unsup_multilingual_mt.

## 1 Introduction

Fully unsupervised machine translation (UMT), where absolutely only unlabeled monolingual corpora are used in the entire model training process, has gained significant traction in recent years. Early success in UMT has been built on the foundation of cross-lingual initialization and iterative back-translation [11, 13, 6, 25, 18]. While achieving outstanding performance on empirically high-resource languages, such as German and French, these UMT methods fail to produce any meaningful translation (near zero BLEU scores) when adopted in low-resource cases, like the FLoRes Nepali and Sinhala unsupervised translation tasks [10]. Arguably, such low-resource languages need UMT the most.

Liu et al. [17] discovered that low-resource UMT models benefit significantly when they are initialized with a multilingual generative model (mBART), which is pre-trained on massive monolingual corpora from 25 languages (CC25). Not only does this multilingual pre-training set contain data of the low-resource languages of interest (*e.g.,* Nepali), it also comprises sentences from related languages (*e.g.,* Hindi), which presumably boosts the performance of their low-resource siblings. Tran et al. [26] and Nguyen et al. [19] later propose ways to mine pseudo-parallel data from multiple directions to improve low-resource UMT. On the other hand, Conneau et al. [7] find the *curse of multilinguality*, which states that performance of a multilingual model with a fixed model capacity tends to decline after a point as the number of languages in the training data increases. Thus, adding more languages may hinder further improvement in low-resource UMT, which is intuitively due to incompatibility between the linguistic structures of different languages. Furthermore, these models have to tell languages apart by using only a single-token language specifier prepended into the input sequence,

which is shown to be insufficient [1]. Our analysis also shows that the models sometimes predict words of the wrong language (see Appendix). Meanwhile, as shown later in our analyses (§4.4), other complementary techniques such as pseudo-parallel data mining may have reached their limits in improving low-resource UMT.

To alleviate the curse of multilinguality and work around the linguistic incompatibility issue in low-resource UMT, we propose a simple refinement strategy aimed at disentangling, or separating, irrelevant languages from a pre-trained multilingual unsupervised model, namely CRISS [26], and focusing exclusively on a number of low-resource UMT directions. Briefly, our method consists of four stages. In the first stage, we use a modified back-translation technique [23] to finetune an initial pre-trained multilingual UMT model on English (En) and a family of low-resource languages $\mathcal{L}$ with a different set of feed-forward layers in the decoder for each $En \leftrightarrow \mathcal{L}$ pair. This step aims at discarding irrelevant languages and separating the languages $\mathcal{L}$ from each other. The second stage separates the resulting $En \leftrightarrow \mathcal{L}$ model into two $En \rightarrow \mathcal{L}$ and $\mathcal{L} \rightarrow En$ models. This stage is motivated by the fact that English is often vastly distinct from low-resource languages $\mathcal{L}$, thus requiring a greater degree of disentanglement and less parameter sharing. The third stage boosts the performance by using the second-stage models as fixed translators for the back-translation process. The final stage, which is only applied to from-English directions, is aimed to separate the target low-resource languages between themselves into different models. Overall, our method prioritises to maximize the individual performance of each low-resource task, though the trade-off is that it can no longer translate multiple languages in a one-for-all manner.

In our experiments, our method establishes the state of the art in fully unsupervised translation tasks of English (En) to Nepali (Ne), Sinhala (Si), Gujarati (Gu), Latvian (Lv), Estonian (Et) and Kazakh (Kk) with BLEU scores of 9.0, 9.5, 17.5, 18.5, 21.0 and 10.0 respectively, and vice versa. This is up to 4.5 BLEU improvement from the previous state of the art [26]. We also show that the method outperforms other related alternatives that attempt to achieve language separation in various low-resource unsupervised tasks. Plus, our ablation analyses demonstrate the importance of different stages of our method, especially the English separation stage (stage 2).

## 2 Background

Recent advances in fully unsupervised machine translation (UMT) are often built on the foundation of iterative back-translation [23, 11, 3, 13]. It is a clever technique where the model back-translates monolingual data from one language to another, and the resulting pair is used to train the model itself via standard back-propagation. To make UMT successfully work, iterative back-translation must be accompanied with some forms of cross- or multi-lingual initialization of the model, either through an unsupervised bilingual dictionary [12, 11], phrase-based statistical MT [13], or language model pre-training [6, 25, 17]. Apart from that, cross-model back-translated distillation [18], where two distinct models are used to complement the target model, can also boost UMT performance. Plus, pseudo-parallel data mining, where sentences from two monolingual corpora are paired to form training samples through certain language-agnostic representations [28, 26, 19], has been shown to significantly improve UMT performance.

Nonetheless, when it comes to fully unsupervised translation of distant and low-resource languages [10], the aforementioned techniques fail to translate their success in high-resource languages to low-resource counterparts, unless "multilinguality" is involved. mBART [17] is among the first multilingual model to boost the performance greatly in low-resource UMT, by pre-training the model on the CC25 dataset, which is a 1.4-terabyte collection of monolingual data from 25 languages. The CC25 dataset contains not only data from English and the low-resource languages of interest, like Nepali, but also data from their related but higher-resource siblings, like Hindi.

CRISS [26] later advances the state of the art in low-resource UMT by finetuning mBART with pseudo-parallel data of more than 180 directions mined from the model's encoder representations. This type of UMT models handles translations in multiple directions by a few principles that are designed to maximize parameter sharing. First, it builds a large shared vocabulary that covers wordpieces from all languages. Second, the encoder encodes input sentences without knowledge about their corresponding languages to promote language-agnostic representations in its outputs. Lastly, the decoder receives the encoder outputs and decodes the translation in the target language by prepending a language-specific token at the beginning of the output sequence. Training the model

in such a multilingual environment helps the encoder learn language-agnostic latent representations that are shared across multiple languages, allowing the decoder to translate from any language. The vast availability of high-resource siblings (*e.g.,* Hindi for Indic languages) may help improve the performance of low-resource MT significantly [9, 5]. Meanwhile, LAgSwAV [19] improve unsupervised MT by mining pseudo-parallel data from monolingual data with cluster assignments.

Despite its outstanding success, multilinguality may conversely hinder the model's further improvement in low-resource unsupervised translation due to the curse of multilinguality [1, 7]. A multilingual model with a fixed capacity may perform suboptimally on individual languages as it is forced to simultaneously handle conflicting structural reorderings of distant languages [16]. The single-token language specifier is also not enough to ensure the language consistency of its predictions [1]. Our proposed method aims to gradually separate the languages and focus on the target low-resource directions only. One prominent feature of our method is that it prioritises to separate English from the remaining low-resource languages first, as these languages are often much more distant from the common language English than from themselves [27].

There are several efforts with similar objectives as ours, albeit of different contexts and scenarios. Specifically, Sen et al. [22] propose a shared encoder and language-specific decoder model to be trained with back-translation for unsupervised MT, which is slightly similar to stage 1 of our proposed four-stage refinement process, as explained in §3. Our stages 2, 3 and 4 improve the first stage, or equivalently Sen et al. [22], further by refining the model in ways that are not possible in stage 1. Meanwhile, Sachan and Neubig [21], Zhang et al. [29], Li and Gong [16] propose methods to cause the model to implicitly "select" relevant language-specific parameters by gating mechanisms or gradient readjustments, in the context of supervised multilingual translation. However, these methods, as found in §4, struggle with unsupervised MT as signals from back-translated synthetic training samples are often much noisier. On the implementation side, the design of our language-specific FFN layers (§3.1) is also largely inspired by sparsely-distributed mixture of experts [24, 14, 30].

Low-resource MT can also benefit greatly from the presence of supervised parallel data, or auxiliary data, in a related language (*e.g.,* English-Hindi for English-Nepali), which is not a fully unsupervised setup but a zero-shot scenario. Garcia et al. [8, 9] propose multi-stage training procedures that combine both monolingual data and auxiliary parallel data to improve the performance of low-resource translation. Meanwhile, Chen et al. [4, 5] suggest to scale their experiments up to 100 languages to attain better performance boosts. Nonetheless, while such line of work offers considerable insight into low-resource MT, it is not within the scope of this paper, as we focus on fully unsupervised machine translation where absolutely no parallel data is available, regardless of languages.

# 3 Language disentanglement refinement

In this section, we describe our multilingual language disentanglement refinement procedure that consists of four stages. Before the training process starts, we first choose to restrict the model, *e.g.,* CRISS [26], to only translate English (En) to (and from) a small group $\mathcal{L} = (l_1, l_2, ..., l_N)$ of $N$ low-resource languages based on their genealogical, demographic, areal or linguistic similarities, such as Nepali (Ne), Sinhala (Si) and Gujarati (Gu) for Indic family (a part of Indo-European). There is no guarantee, however, that these languages are indeed significantly close due to such heuristics. In the following, we first propose a language-specific feed-forward layer (§3.1), which helps achieve a gradual language separation process (Figure 1). Then, in §3.2-§3.5, we explain the four stages of our language separation procedure, which is also visually demonstrated in Figure 2.

## 3.1 Language-specific sharded feed-forward layer

While our method starts with finetuning a fully parameter-shared multilingual MT model, we desire to gradually and progressively increase the level of language separation by reducing the degree of parameter sharing of the model. Thus, for every layer $j$ of an $H$-layer Transformer decoder such that $j$ is divisible by a constant $\sigma$ (*e.g.,* 3) and $1 \leq j \leq H$, we propose to replace its vanilla feed-forward layer (FFN$_j$) with $N$ separate layers $\overline{\text{FFN}}_{j,i}$ such that each corresponds to a language $l_i \in \mathcal{L}$.

During the training process, each of the separate $\overline{\text{FFN}}_{j,i}$ layers is sharded and reside individually on a distinct GPU-$i$ accelerator, where their back-propagated gradients are not aggregated or averaged during the data parallelism process. Each GPU-$i$ device is only fed with data from its

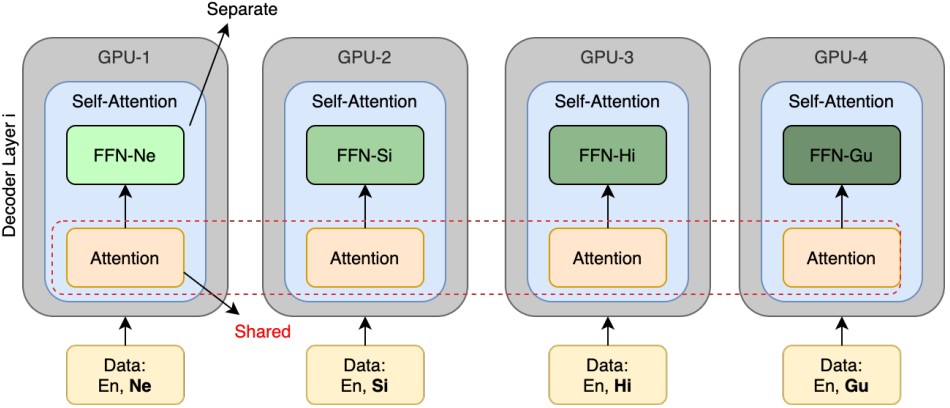

Figure 1: Illustration of a decoder layer with language-specific sharded FFN layers, where the FFN parameters for each language pair are separate and reside on a specific GPU and only data streams from such language pair (*e.g.,* En, Ne for GPU-1) are fed into the model replica in that GPU.

respective language $l_i$ and the common language English. Meanwhile, the remaining parameters (*e.g.,* embedding and attention layers) are still shared and their gradients are averaged across all GPU devices during back-propagation. Figure 1 visualizes a sharded FFN layer of our model and how its parameters are allocated on each GPU. During inference, $\overline{\text{FFN}}_{j,i}$ is used to decode the translation to and from $l_i$.

Formally, let $\hat{\theta} = \{\hat{\theta}_m, \hat{\theta}_f\}$ be a fully parameter-shared multilingual model (*e.g.,* CRISS) at our initialization, where $\hat{\theta}_m$ is the set of parameters intended to be shared and $\hat{\theta}_f$ is the initial set of FFN parameters intended to be disentangled in our proposed model. We define our model as $\theta = \{\theta_m, \theta_f^1, \theta_f^2, ..., \theta_f^N\}$, where $\theta_m$ denotes the shared parameters while $\theta_f^i$ denotes the separate parameters of all $\overline{\text{FFN}}_{j,i}$ layers ($1 \leq j \leq H$) for a language $l_i \in \mathcal{L}$. Before we begin training the model, we initialize $\theta_m = \hat{\theta}_m$ and $\theta_f^i = \hat{\theta}_f$, $\forall i \in [1, ..., N]$. Then during each update step, the gradients of $\theta_f^i$ and $\theta_m$ are computed as follow:

$$\nabla_\theta = \{\nabla_{\theta_m}, \nabla_{\theta_f^1}, \ldots, \nabla_{\theta_f^N}\} \tag{1}$$

$$\nabla_{\theta_f^i} = \mathop{\mathbb{E}}_{\substack{x_i \sim \mathbb{X}_i \\ x_e \sim \mathbb{X}_e}} \nabla_{\theta_f^i} \big(\mathcal{J}_\theta(x_i|y_i^e) + \mathcal{J}_\theta(x_e|y_e^i)\big) \tag{2}$$

$$\nabla_{\theta_m} = \frac{1}{N} \sum_{l_i \in \mathcal{L}} \mathop{\mathbb{E}}_{\substack{x_i \sim \mathbb{X}_i \\ x_e \sim \mathbb{X}_e}} \nabla_{\theta_m} \big(\mathcal{J}_\theta(x_i|y_i^e) + \mathcal{J}_\theta(x_e|y_e^i)\big) \tag{3}$$

where $\mathbb{X}_i$ and $\mathbb{X}_e$ are monolingual corpora (or data distribution) of language $l_i$ and English respectively, $\mathcal{J}_\theta(x|y) = -\log P_\theta(x|y)$, and $y_i^e \sim P(\cdot|x_i, \{\theta_m, \theta_f^i\})$ is the back-translation into English from $x_i$ while $y_e^i \sim P(\cdot|x_e, \{\theta_m, \theta_f^i\})$ is the back-translation into $l_i$ from English by the model $\{\theta_m, \theta_f^i\}$. Equations 2 and 3 show that gradients are not aggregated for the language-specific sharded FFN layers, while the remaining are aggregated and averaged across all data streams. However, note that our model only separates the FFN layers of the decoder, while all encoder parameters are fully shared to ensure the language-agnostic representations from the encoder [26, 19].

**Implementation aspect.** The rationale behind the design of our language-specific sharded FFNs is to take full advantage of multi-GPU environment to maximize training efficiency both in speed and batch size. First, each sharded FFN is placed on a separate GPU and only the respective language-specific data streams are fed into that GPU, which enables us to achieve the same maximal batch size as the regular Transformer without running into out-of-memory error. Second, FFN layers, rather than other types of layers (*e.g.,* attention), are chosen to be separated because they require minimal additional memory (only 50Mb for a 8.8Gb model) to achieve noticeable performance gain (see §4.2). If we have access to only one GPU, we can allocate all FFNs into it, reduce the batch size and increase gradient accumulation to achieve the same multi-GPU effect.

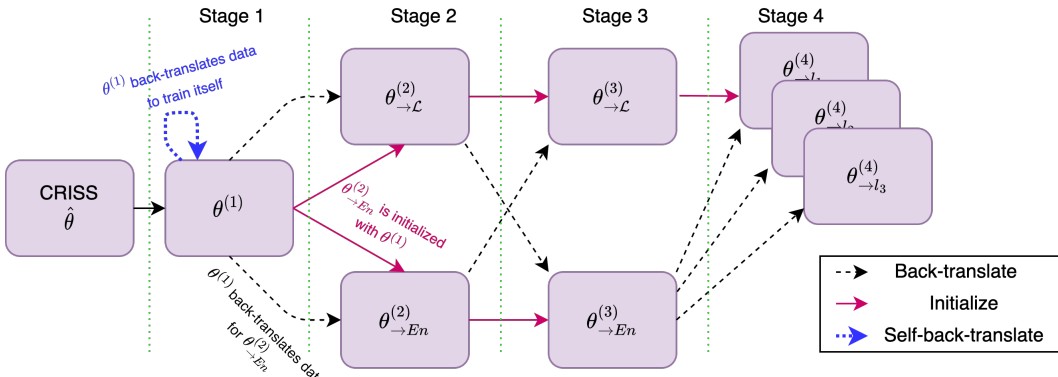

Figure 2: Overview of the stages of our language disentanglement refinement procedure. Stage 1 finetunes our model with language-specific FFNs using multilingual back-translation. Then, stage 2 and stage 3 continue by separating the model into to-En and from-En models. Finally, stage 4 separates low-resource target language from the resulting from-En model.

## 3.2 First stage

With the aforementioned language-specific FFN layers, we enter the first stage of refinement. The purpose of this stage is to begin differentiating the language-specific FFN layers so that each FFN can specialize in its designated language. That is, after initializing our model $\theta = \{\theta_m, \theta_f^1, ..., \theta_f^N\}$ with the pre-trained multilingual UMT model $\hat{\theta} = \{\hat{\theta}_m, \hat{\theta}_f\}$ as $\theta_m = \hat{\theta}_m$ and $\theta_f^i = \hat{\theta}_f \ \forall i \in [1, ..., N]$, we finetune it with iterative back-translation [6], except with some noteworthy modifications. Specifically, we uniformly sample monolingual sentences $x_e$ and $x_i$ from English and each low-resource language $l_i \in \mathcal{L}$, respectively. Then, we use $\theta_i = \{\theta_m, \theta_f^i\}$, a subset of $\theta$, to back-translate $x_e$ and $x_i$ into $y_e^i$ and $y_i^e$, respectively. After that, we train the model with $y_e^i \to x_e$ and $y_i^e \to x_i$ pairs. Note that for each direction of back-translation, forward and backward directions, the appropriate FFN layers $\theta_f^i$ are used with respect to the data stream involving language $l_i$ and English on a separate GPU-$i$. Furthermore, similarly to Conneau and Lample [6], the back-translation model is continuously updated after each training step. The resulting model $\theta^{(1)} = \{\theta_m^{(1)}, \theta_f^{1,(1)}, ..., \theta_f^{N,(1)}\}$ will be used to initialize the second stage of refinement.

## 3.3 Second stage

Alleviating the curse of multilinguality may involve separation of unrelated languages. Yet, for many low-resource translation tasks, English is often significantly distant from the target low-resource counterparts, such as Nepali and Sinhala, which in fact share certain similarities between themselves. Thus, English may not share similar structural patterns with any of the target languages [27]. Despite that, most existing UMT models are bidirectional [6, 25, 18]. This means that they have to handle both translations to English and their target low-resource language $l_i$, causing them to endure reordering discrepancy and perform sub-optimally for both directions.

The second stage is aimed to *disentangle English*. It builds two separate models $\theta_{\to\mathcal{L}}^{(2)}$ and $\theta_{\to En}^{(2)}$ from $\theta^{(1)}$, which are specialised to translate to target low-resource languages $l_i \in \mathcal{L}$ and to English, respectively. Specifically, the second stage starts by initializing both $\theta_{\to\mathcal{L}}^{(2)}$ and $\theta_{\to En}^{(2)}$ with $\theta^{(1)}$. We then finetune them with iterative back-translation exclusively in $En \to l_i$ and $l_i \to En$ directions, respectively. Different from the first stage, we use a fixed model $\theta^{(1)}$ to back-translate the monolingual data. In other words, to finetune $\theta_{\to\mathcal{L}}^{(2)}$, we first initialize it with $\theta^{(1)}$. Then we sample monolingual data $x_i$ uniformly from all $l_i \in \mathcal{L}$ and use the fixed model $\theta^{(1)}$ to back-translate them into English $y_i^e \sim P(\cdot|x_i, \{\theta_m^{(1)}, \theta_f^{i,(1)}\})$. The resulting pair $y_i^e \to x_i$ is used to train $\theta_{\to\mathcal{L}}^{(2)}$. In the opposite direction, we finetune $\theta_{\to En}^{(2)}$ by first initializing it with $\theta^{(1)}$, then sample English data $x_e$ and

randomly choose an $l_i \in \mathcal{L}$ to back-translate $x_e$ into $y_e^i \sim P(\cdot|x_e, \{\theta_m^{(1)}, \theta_f^{i,(1)}\})$, and finally train $\theta_{\rightarrow En}^{(2)}$ with $y_e^i \rightarrow x_e$ pairs.

## 3.4 Third stage

The third stage improves the performance further by continuing finetuning $\theta_{\rightarrow \mathcal{L}}^{(2)}$ and $\theta_{\rightarrow En}^{(2)}$ into $\theta_{\rightarrow \mathcal{L}}^{(3)}$ and $\theta_{\rightarrow En}^{(3)}$, respectively. Specifically, we build $\theta_{\rightarrow \mathcal{L}}^{(3)}$ by initializing it with $\theta_{\rightarrow \mathcal{L}}^{(2)}$ and use $\theta_{\rightarrow En}^{(2)}$ as a fixed back-translation model and train $\theta_{\rightarrow \mathcal{L}}^{(3)}$ in the same manner as in §3.3. Similarly, $\theta_{\rightarrow En}^{(3)}$ is trained in the same fashion in the opposite direction.

## 3.5 Fourth stage

The last stage is aimed to achieve the maximal degree of language separation not only from English, but also between all languages in the target group. Specifically, we seek to train $N$ separate models $\theta_{\rightarrow l_i}^{(4)}$ for each language $l_i \in \mathcal{L}$. Each model $\theta_{\rightarrow l_i}^{(4)}$ is initialized with $\theta_{\rightarrow \mathcal{L}}^{(3)}$ with the corresponding language-specific FFN layers, *i.e.*, $\{\theta_m^{(3)}, \theta_f^{i,(3)}\}_{\rightarrow \mathcal{L}}$, and $\theta_{\rightarrow l_i}^{(4)}$ is trained by back-translation with a fixed backward model $\theta_{\rightarrow En}^{(3)}$, and exclusively on $En \rightarrow l_i$ direction. Because only one direction is trained in this stage, we no longer use the above language-specific FFN layers for our end models. Instead, we revert back to the vanilla Transformer, whose FFN layers are initialized with the appropriate language-specific layers from the previous stage. Note that we do not perform this stage of refinement for the to-English ($\rightarrow En$) direction, due to the fact that the encoder is fully shared across all languages and there is no difference in separating $l_i \rightarrow En$ directions for the encoder, while the decoder only decodes the common English language.

# 4 Experiments

In this section, we tested our method on the standard low-resource unsupervised machine translation (§4.1). We then empirically analyze various aspects of our method and compare it with other possible variants (§4.2) as well as other related alternatives in the literature (§4.3).

## 4.1 Low-resource unsupervised translation

**Setup.** We evaluate our method on the FLoRes [10] low-resource unsupervised translation tasks on English (En) to and from Nepali (Ne), Sinhala (Si) and Gujarati (Gu) from the Indic family, as well as other low-resource languages: Latvian (Lv) and Estonian (Et) from the Uralic family and Kazakh (Kk) from the Turkic family. Although Hindi (Hi) and Finnish (Fi) are relatively high-resourced compared to their respective siblings, we still add them into the mix for Indic and Uralic families respectively to help the learning process of their respective low-resource siblings, following [9, 5].

We start finetuning our model using the exact same codebase and the published pre-trained CRISS model [26]. We also use the same sentencepiece pre-processing and vocabulary. Specifically, we group the tested languages into 3 groups: Indic (En-{Ne,Si,Hi,Gu}), Uralic (En-{Fi,Lv,Et}) and Turkic (En-Kk). For each group, we use monolingual corpora from the relevant languages of the Common Crawl dataset, which contains data from totally 25 languages [17]. We limit the amount of monolingual data per language to up to 100M sentences. Similarly to CRISS, our model is a 12-layer Transformer, whose decoder's FFN layers are replaced with our language-specific sharded FFN layers with $\sigma = 3$ (see §3.1) during the first 3 stages. For each group of $N$ languages, we use $N$ GPUs to train the model and shard each language-specific FFN layer to a single GPU, where we channel only data streams of the respective languages, according to our formulations in §3.2. We use a batch size of 1024 tokens with a gradient accumulation factor of 2 [20]. Similarly to the baselines [26, 19], we use the multi-bleu.perl[1] to evaluate the BLEU scores with a beam size of 5. In each stage, we finetune the models for 20K updates. Further details, such as corresponding SacreBLEU scores for reference, are provided in the Appendix.

---

[1] To ensure the scores are comparable, we matched the entire evaluation pipeline (test sets, tokenzation, scripts, etc) with the baselines [26, 19] and consistently reproduced baselines' results with our pipeline.

Table 1: Comparison of BLEU scores for different methods on fully unsupervised translation tasks of various low-resource languages from the Indic, Uralic and Turkic language families.

| Method | Indic | | | | | | | | Uralic | | | | | | Turkic | |
|---|---|---|---|---|---|---|---|---|---|---|---|---|---|---|---|---|
| | En-Ne | Ne-En | En-Si | Si-En | En-Hi | Hi-En | En-Gu | Gu-En | En-Fi | Fi-En | En-Lv | Lv-En | En-Et | Et-En | En-Kk | Kk-En |
| mBART | 4.4 | 10.0 | 3.9 | 8.2 | - | - | - | - | - | - | - | - | - | - | - | - |
| LAgSwAV | 5.3 | 12.8 | 5.4 | 9.4 | - | - | - | - | - | - | - | - | - | - | - | - |
| CRISS | 5.5 | 14.5 | 6.0 | 14.5 | 19.4 | 23.6 | 14.2 | 23.7 | 20.2 | 26.7 | 14.4 | 19.2 | 16.8 | 25.0 | 6.7 | 14.5 |
| Ours | **9.0** | **18.2** | **9.5** | **15.3** | **20.8** | **23.8** | **17.5** | **29.5** | **22.9** | **28.2** | **18.5** | **19.3** | **21.0** | **25.7** | **10.0** | **20.0** |
| $+\Delta$ | 3.5 | 3.7 | 3.5 | 0.8 | 1.4 | 0.2 | 3.3 | 5.8 | 2.7 | 1.5 | 4.1 | 0.1 | 4.2 | 0.7 | 3.3 | 5.5 |

Table 2: Stage-wise ablations of our method: BLEU scores for different refinement stages on the fully unsupervised translation tasks of various low-resource languages from the Indic, Uralic and Turkic language families.

| Method | Indic | | | | | | Uralic | | | | Turkic | |
|---|---|---|---|---|---|---|---|---|---|---|---|---|
| | En-Ne | Ne-En | En-Si | Si-En | En-Gu | Gu-En | En-Lv | Lv-En | En-Et | Et-En | En-Kk | Kk-En |
| CRISS | 5.5 | 14.5 | 6.0 | 14.5 | 14.2 | 23.7 | 14.4 | 19.2 | 16.8 | 25.0 | 6.7 | 14.5 |
| +Stage 1 | 7.8 | 16.9 | 7.1 | 14.5 | 15.8 | 26.2 | 15.2 | 19.0 | 17.4 | 24.9 | 7.9 | 16.9 |
| +Stage 2 | 8.6 | 17.7 | 8.7 | 15.0 | 16.4 | 28.8 | 16.3 | 19.2 | 19.7 | 25.6 | 9.2 | 18.5 |
| +Stage 3 | 8.8 | 18.2 | 9.0 | 15.3 | 16.8 | 29.5 | 17.9 | 19.3 | 20.0 | 25.7 | 9.7 | 20.0 |
| +Stage 4 | 9.0 | 18.3 | 9.5 | 15.3 | 17.5 | 29.1 | 18.5 | 19.2 | 21.0 | 25.7 | 10.0 | 19.9 |

**Results.** Table 1 shows the performances of our method against recent best baselines, namely mBART [17], LAgSwAV [19] and CRISS [26], on the fully unsupervised translation tasks between English and the low-resource languages from the Indic, Uralic and Turkic families. As shown, our method achieves the state of the art in En-Ne and En-Si FLoRes low-resource UMT tasks, with 9.0 and 9.5 BLEU, as well as other languages across the three families, such as 21.0 and 10.0 BLEU for En-Et and En-Kk tasks. It surpasses the previous best [26] by up to 5.8 BLEU and achieves a mean gain of 2.6 BLEU across the 16 directions. The results also indicate that our method achieves more gain margin for low-resource languages (Ne, Si, Gu) compared to their high-resource siblings (Hi).

## 4.2 Ablation study

In this section, we conduct a comprehensive ablation study to gain more insights of our method and the importance of its components. In particular, we examine the contribution of each stage of our procedure to the overall performance gain across various low-resource languages (Table 2). We also investigate how our decoder language-specific FFN layers perform in comparison to other possible variants, such as language-specific attention layers.

**Stage-wise ablation study.** Table 2 shows the performances at different refinement stages of our method in low-resource fully unsupervised translation between English to and from Nepali, Sinhala, Gujarati, Latvian, Estonian and Kazakh. As it can be seen, our first and second stages achieve significant kick-off gain from the CRISS baseline, with up to 5.1 BLEU score for Gu-En direction. We also observe considerable improvement from stage 1 to stage 2, with up to 2.7 BLEU. This indicates that separation of English from other low-resource languages play a crucial role. Meanwhile, the third and fourth stages improve the performances further, though with less margin. In addition, it is noteworthy that stage 4 does not noticeably outperforms stage 3 for to-English ($\rightarrow En$) direction, due to the fact that the encoder is fully shared across all low-resource languages and there is no difference in disentangling $l_i \rightarrow En$ directions from the encoder's perspective. Thus, there is no point training stage 4 for to-English direction, as formulated in our stage 4 (see §3.5).

**Language-specific variants.** Other than our decoder language-specific FFN layers explained in §3.1, there could be other possible ways to shard model parameters into language-specific components. We compare some of these variants against our own in Table 3 to examine their effects on the FLoRes low-resource unsupervised translation tasks with Nepali, Sinhala and Gujarati. Specifically, we compare our method (Lg. FFN/Dec) with a similar variant *without* the language-specific FFN layers (w/o Lg. FFN/Dec), with decoder language-specific self- and cross-attention layers (Lg. Att/Dec), with language-specific FFN layers in the **encoder** (Lg. FFN/Enc), and finally with language-specific FFN layers in both encoder and decoder (Lg. FFN/Enc+Dec). To ensure consistency, we only show results up to stage 3 of our procedure and skip the fourth stage, where language-specific elements

Table 3: BLEU scores for different variants of our method in the Indic languages: without decoder language-specific (LS) FFN (w/o Lg. FFN/Dec), with decoder LS attention layers (Lg. Att/Dec) instead, with LS FFNs in encoder instead of decoder (Lg. FFN/Enc), with LS FFNs in both encoder and decoder (Lg. FFN/Enc+Dec). Results reported after stage 3 of our refinement procedure because stage 4 no longer contains language-specific elements.

| Method | Mem | En-Ne | Ne-En | En-Si | Si-En | En-Gu | Gu-En |
|---|---|---|---|---|---|---|---|
| CRISS | 0% | 5.5 | 14.5 | 6.0 | 14.5 | 14.2 | 23.7 |
| Ours (Lg. FFN/Dec) | 4% | 8.8 | 18.2 | 9.0 | 15.3 | 16.8 | 29.5 |
| w/o Lg. FFN/Dec | 0% | 7.9 | 17.3 | 8.3 | 14.7 | 15.7 | 27.8 |
| Lg. Att/Dec | 4% | 8.4 | 17.7 | 8.6 | 15.0 | 15.7 | 28.7 |
| Lg. FFN/Enc | 4% | 6.5 | 15.1 | 7.1 | 13.9 | 14.9 | 26.0 |
| Lg. FFN/Enc+Dec | 8% | 8.9 | 18.2 | 8.8 | 15.3 | 17.0 | 29.1 |

no longer apply. In addition to the BLEU scores, we also report the percentage of excess memory overhead needed for each method compared to the baseline CRISS (0% overhead).

As shown in Table 3, the variant without any language-specific parameters (w/o Lg.FFN/dec) underperforms our method by up to 1.0 BLEU, even though English is separated in stage 2 and stage 3. This indicates that it is essential for the method to have a gradual language disentanglement process. Furthermore, placing the language-specific FFN layers on the encoder only (Lg. FFN/Enc) causes even more detrimental performance drop. The results imply that it is not advisable to linguistically separate the encoder only. Similarly, having language-specific parameters on the attention layers (Lg. Att/Dec) does not perform as good as our FFN counterpart, despite having the same memory footprint overhead of 4%. Meanwhile, placing language-specific FFN layers on both encoder and decoder (Lg. FFN/Enc+Dec) yields relatively similar results as our own method, although it requires double the memory overhead. Thus, our choice of separate FFN parameters on the decoder only is more favorable given its outperforming capability while consuming less memory. The Appendix provides further ablation analysis this issue.

## 4.3   Comparison with related methods

In this section, we compare our refinement procedure against other related methods, as well as those that share a similar language disentanglement objective as ours. Particularly, we reproduce the method proposed by Sen et al. [22] in low-resource unsupervised MT. This method is equivalent to stage 1 of our refinement procedure, except that the entire decoder is linguistically disentangled with separate decoders for different languages, not just FFN parameters. Meanwhile, Sachan and Neubig [21] and Zhang et al. [29] both share the same goal of alleviating the curse of multilinguality, but in the context of supervised multilingual translation, where the model teaches itself to implicitly "select" relevant language-specific layers by gating mechanisms or gradient readjustments. We adapt these methods to unsupervised MT by substituting supervised training data with multilingual back-translated samples by the models. Lastly, Zuo et al. [30] recently propose a mixture-of-experts (MoE) variant [14, 24], where experts (FFNs) are selected randomly to perform the forward pass. In the context of language disentanglement, these separate experts act as implicit language-specific components. We also adapt this method to low-resource unsupervised tasks with multilingual back-translation and set the number of experts as the number of languages. For all methods, we use CRISS as the initial model.

The comparison results are shown in Table 4. As shown, the method proposed by Sen et al. [22] performs slightly above stage 1 of our method, while it underperforms our complete procedure by up to 2.7 BLEU in Gu-En task. This is in line with the fact that it is almost equivalent to our first stage. The approaches suggested by Sachan and Neubig [21] and Zhang et al. [29], on the other hand, yield considerably lower BLEU scores in various low-resource unsupervised tasks. A possible reason for this underperformance is that these methods are designed for supervised multilingual translation, where accurate parallel data plays a crucial role. When adapting to the unsupervised regime, the synthetic training samples created by back-translation may be too noisy and inaccurate. Meanwhile, the MoE candidate [30] also produce lower BLEU scores than our method, which can be due to its lack of explicit language-specific components to enforce a language disentanglement naturally.

Table 4: Comparison with related methods. * indicates the cited method is exclusively applied to supervised MT, and it is adapted to unsupervised MT using multilingual iterative back-translation products as training samples.

| Method | En-Ne | Ne-En | En-Si | Si-En | En-Gu | Gu-En |
|---|---|---|---|---|---|---|
| CRISS | 5.5 | 14.5 | 6.0 | 14.5 | 14.2 | 23.7 |
| **CRISS finetuned with** | | | | | | |
| Sen et al. [22] | 7.7 | 17.0 | 7.0 | 14.5 | 15.7 | 26.8 |
| Sachan and Neubig [21]* | 7.1 | 16.9 | 7.2 | 14.0 | 15.1 | 26.1 |
| Zhang et al. [29]* | 6.8 | 16.1 | 6.4 | 14.0 | 14.9 | 25.2 |
| Zuo et al. [30]* | 8.2 | 17.4 | 7.8 | 14.6 | 16.4 | 25.6 |
| Ours | 9.0 | 18.2 | 9.5 | 15.3 | 17.5 | 29.5 |

Table 5: Comparison with popular UMT techniques.

| Method | En-Ne | Ne-En | En-Si | Si-En | En-Gu | Gu-En |
|---|---|---|---|---|---|---|
| CRISS | 5.5 | 14.5 | 6.0 | 14.5 | 14.2 | 23.7 |
| **CRISS finetuned with** | | | | | | |
| CBD | 7.6 | 16.9 | 7.3 | 14.6 | 15.8 | 26.0 |
| Mined data | 4.6 | 10.5 | 4.8 | 9.8 | 11.2 | 19.2 |
| Mined data + BT | 6.6 | 16.1 | 6.7 | 13.1 | 14.7 | 24.5 |
| Ours | 9.0 | 18.2 | 9.5 | 15.3 | 17.5 | 29.5 |

Table 6: Back-translated pseudo-parallel dataset sizes.

| Mined data size | En-Ne | En-Si |
|---|---|---|
| Unfiltered | ~109.8M | ~108.8M |
| Filtered | ~3000 | ~6000 |

## 4.4 Limit of other unsupervised techniques

Apart from iterative back-translation, there are other techniques that have been shown to improve unsupervised machine translation. One of them is cross-model back-translated distillation [18], or CBD, where two distinct UMT teachers are used to distill the final model. Another is pseudo-parallel data mining [26, 19], where language-agnostic representations are built for sentences across different languages, which are then used to map unlabeled sentences from one language to another to create synthetic parallel data [2]. CRISS is also itself pseudo-parallel data mining method. In this segment, we demonstrate that the aforementioned techniques do not adapt well with low-resource unsupervised tasks, which shifts our attention to other areas, like the curse of multilinguality. Specifically, we apply CBD on CRISS by finetuning two distinct models from pre-trained CRISS and using them to distill a final model. For pseudo-parallel mining, we use LASER [2] to mine pseudo-parallel data in only the low-resource directions of interest using the encoder outputs from CRISS itself.

The results are reported in Table 5. As shown, CBD only performs relatively similar to our stage 1. This indicates that the CBD only improves due to back-translation, and not its own distillation effect. Nonetheless, CBD faces a disadvantage that the two teachers are not considerably distinct as they are both finetuned from the same pre-trained CRISS model, which is not advisable in the original paper. This is nonideal as there is no existing well-performing pre-trained model other than CRISS.

Meanwhile, the mined pseudo-parallel data contributes little to the performance improvement due to the reality that the amount and quality of mined data is too low for low-resource pairs, which is less than 5000 pairs for En-Ne and En-Si. This forces the model to be trained on such small mined datasets by progressively decreasing loss weight to prevent overfitting. While many may attribute this to the model's failure in mining more high-quality pseudo-parallel data, we empirically show that it is more likely that *there are in fact not enough real parallel data in low-resource corpora to be mined!* Specifically, we use our outperfoming model to back-translate the entire English monolingual corpus to Nepali. For each resulting back-translated Ne sentence, we search for its closest real Ne sentence by token-based Levenshtein distance [15] in the Ne monolingual corpus, and then filter out pairs whose distance is more than 20% the sentence length. We repeat the process for En-Si as well. As shown in Table 6, out of >100M possible unfiltered pairs, only <6000 samples satisfy the filtering criterion. We provide an in-depth analysis on this issue in the Appendix.

## 5 Conclusion

We proposed a simple refinement procedure to alleviate the curse of multilinguality in low-resource unsupervised machine translation. Our method achieves the state of the art in the FLoRes low-resource task of English-Nepali and English-Sinhala, as well as other language low-resource pairs across 10 language directions in Indic, Uralic and Turkic families. Further analyses show that the method outperforms other related alternatives when adapting to low-resource unsupervised translation.

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
