# Appendix: Refining Low-Resource Unsupervised Translation by Language Disentanglement of Multilingual Model

**Xuan-Phi Nguyen**[1,3], **Shafiq Joty**[1,2], **Wu Kui**[3], **Ai Ti Aw**[3]
[1]Nanyang Technological University
[2]Salesforce Research
[3]Institute for Infocomm Research (I[2]R), A*STAR
Singapore
{nguyenxu002@e.ntu,srjoty@ntu}.edu.sg
{wuk,aaiti}@i2r.a-star.edu.sg

## A   Appendix

In the Appendix, we provide further details about the statistics of the datasets used for training and evaluation, as well as the experimental setups. We also conduct analyses experiments into the reason why further pseudo-parallel data mining fails to improve low-resource unsupervised translation.

### A.1   Statistics of Datasets and Experimental Setups

**Data statistics.**   Table 1 provides details about the amount of monolingual data for each corpus of different languages that are tested in this paper. Apart from English, Hindi, Finnish, the amounts available to be trained for other low-resource languages are particularly low, making improving rather more difficult. Table 2, meanwhile, provide the origins of the test sets used to evaluate the performances of the models on the low-resource tasks. The test set sources are the same as used in CRISS paper [9].

Table 1: Statistics on the sizes of monolingual corpora that are used in the paper.

|                | No. Sentences |
| -------------- | ------------- |
| English (En)   | 100M          |
| Nepali (Ne)    | 9.8M          |
| Sinhala (Si)   | 8.8M          |
| Hindi (Hi)     | 80M           |
| Gujarati (Gu)  | 5.6M          |
| Finnish (Fi)   | 61M           |
| Latvian (Lv)   | 35M           |
| Estonian (Et)  | 26M           |
| Kazakh (Kk)    | 33M           |

**Experimental details.**   For all low-resource experiments, we use the same architectural setups as CRISS [9]. That is, we use a Transformer with 12 layers, dimension of 1024 with 16 attention heads. We finetune the models with cross-entropy loss with 0.2 label smoothing, 0.3 dropout, 2500 warm-up steps. We use a learning rate of 3e-5. When decoding, we use beam size of 5 and length penalty of 0.1 for Indic languages and 0.5 for other languages.

36th Conference on Neural Information Processing Systems (NeurIPS 2022).

Table 2: Test sets used for each language pair.

| | Test set |
|---|---|
| En-Ne | FLoRes [5] |
| En-Si | FLoRes [5] |
| En-Hi | ITTB [6] |
| En-Gu | WMT19 [2] |
| En-Fi | WMT17 [3] |
| En-Lv | WMT17 [3] |
| En-Et | WMT18 [4] |
| En-Kk | WMT19 [2] |

Table 3: Comparison of BLEU scores for different methods on fully unsupervised translation tasks of various low-resource languages from the Indic, Uralic and Turkic language families. SacreBLEU [8] numbers are reported in subscript.

| Method | Indic | | | | | | | | Uralic | | | | | | Turkic | |
|---|---|---|---|---|---|---|---|---|---|---|---|---|---|---|---|---|
| | En-Ne | Ne-En | En-Si | Si-En | En-Hi | Hi-En | En-Gu | Gu-En | En-Fi | Fi-En | En-Lv | Lv-En | En-Et | Et-En | En-Kk | Kk-En |
| mBART | 4.4 | 10.0 | 3.9 | 8.2 | - | - | - | - | - | - | - | - | - | - | - | - |
| LAgSwAV | $5.3_{5.4}$ | $12.8_{12.5}$ | $5.4_{5.3}$ | $9.4_{5.1}$ | - | - | - | - | - | - | - | - | - | - | - | - |
| CRISS | $5.5_{5.6}$ | $14.5_{14.4}$ | $6.0_{6.0}$ | $14.5_{14.3}$ | $19.4_{19.5}$ | $23.6_{23.1}$ | $14.2_{14.2}$ | $23.7_{23.4}$ | $20.2_{20.1}$ | $26.7_{26.2}$ | $14.4_{14.3}$ | $19.2_{18.6}$ | $16.8_{16.7}$ | $25.0_{24.6}$ | $6.7_{6.7}$ | $14.5_{14.5}$ |
| Ours | $9.0_{9.1}$ | $18.2_{18.1}$ | $9.5_{9.4}$ | $15.3_{15.1}$ | $20.8_{20.9}$ | $23.8_{23.4}$ | $17.5_{17.5}$ | $29.5_{29.1}$ | $22.9_{22.8}$ | $28.2_{27.8}$ | $18.5_{18.2}$ | $19.3_{18.9}$ | $21.0_{20.9}$ | $25.7_{25.4}$ | $10.0_{10.0}$ | $20.0_{19.7}$ |
| $+\Delta$ | 3.5 | 3.7 | 3.5 | 0.8 | 1.4 | 0.2 | 3.3 | 5.8 | 2.7 | 1.5 | 4.1 | 0.1 | 4.2 | 0.7 | 3.3 | 5.5 |

## A.2 Low-resource Unsupervised Translation Results in SacreBLEU

Because the compared baselines [9, 7] reported multi-bleu.perl scores, **??** in the main paper is also reported with multi-bleu.perl scores for consistent purpose, which are carefully calibrated to match the exact evaluation pipeline (test sets, tokenization, scripts, pre-trained models, etc) used in the previous work. In this section, we present the corresponding SacreBLEU [8] scores to provide better insights and references. As shown in Table 3, SacreBLEU scores deviate from multi-bleu.perl scores by a negligible amount or up to 0.5 BLEU, depending the tested language pair.

## A.3 Extra Ablation Study

In addition to the ablation study in the main paper, we also conduct further analysis on the impact of the $\sigma$ hyper-parameter, which dictates how many FFN layers in the original Transformer are replaced with our language-specific sharded FFN layers. Specifically, for every layer $j$ of an $H$-layer Transformer decoder that $j$ is divisible by $\sigma$ and $1 \leq j \leq H$, we replace it with our language specific FFN layer. As a result, as $\sigma$ increases, the number of replaced FFNs decreases.

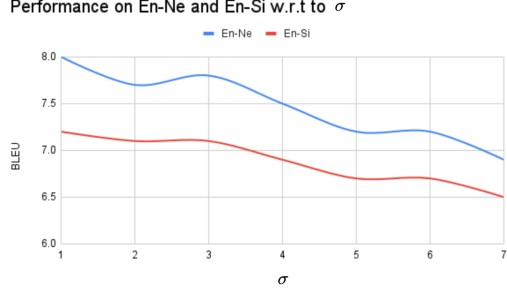

Figure 1: BLEU performance on unsupervised machine translation for En-Ne and En-Si with respect to $\sigma$. In our main experiments, we chose $\sigma = 3$ because it offers relative optimal performance while requires the least extra parameters for stage 1 to 3.

## A.4 Pseudo-parallel Data Mining Analysis

Pseudo-parallel data mining has become a crucial element of unsupervised machine translation [9, 7]. CRISS [9], which we used extensively in our paper, is also itself a pseudo-parallel data mining method. This strategy involves first building a shared multilingual encoder that is able to produce language-agnostic representations of sentences across different languages. This means the model is able to map sentences of similar semantic meanings into the same latent space, regardless of which language they belong to. These latent representations can be used by LASER [1] to find and map monolingual data from two corpora together to create a synthetic parallel dataset.

In this section, given its success, we investigate whether pseudo-parallel mining can further improve the performance in low-resource unsupervised machine translation. Specifically, we use pre-trained CRISS to mine pseudo-parallel data from the monolingual corpora of English against Indic low-resource languages Nepali and Sinhala. Then, we use such data to finetune CRISS for 5000 steps (CRISS + mined data) for each low-resource pair, which is equivalent to how CRISS was originally trained, except with mined data from 180 other directions. Since the size of these mined datasets are small, we also finetune CRISS with them along with iterative back-translation on the target language pairs (CRISS + mined data + BT).

The results are reported in Table 4. Specifically, with back-translation, the mined datasets cause detrimental damage to the model's performance, mainly due to their unbearably small sizes (less than 5000 samples). Further inspections into these mined datasets show that they contains lots of mismatches or incorrect alignments, which adds more noise to the model. When combined with iterative back-translation (CRISS + mined data + BT), the model manages to beat the baseline, although the results indicates that the gains are thanks to back-translation instead of the mined data.

Table 4: Comparison with popular unsupervised MT techniques, such as CBD or CRISS pseudo-parallel data mining.

|  | En-Ne | Ne-En | En-Si | Si-En |
|---|---|---|---|---|
| **Data information** | | | | |
| Corpus size | En: 100M, Ne: 9.8M | | En: 100M, Si: 8.8M | |
| Mined data size | ~4800 | | ~3600 | |
| **Performance** | | | | |
| CRISS | 5.5 | 14.5 | 6.0 | 14.5 |
| CRISS + mined data | 4.6 | 10.5 | 4.8 | 9.8 |
| CRISS + mined data + BT | 6.6 | 16.1 | 6.7 | 13.1 |
| Ours | 9.0 | 18.2 | 9.5 | 15.3 |

Upon seeing the above results, it is natural to doubt that CRISS is simply unable to mine more and better data, and a better language-agnostic encoder can mine more data. However, we empirically found that this may not be the case. To be more precise, we use our outperforming model after the proposed refinement to back-translate all sentences from the English corpus into the target low-resource language, such as Nepali (Ne). Then, for each of the back-translated Ne sentence, we use LASER to search for top 20 nearest neighbors in the real Nepali corpus based on their CRISS embeddings. We then choose the best match by the shortest Levenshtein distance between the back-translated sentence and the real ones. We perform a similar procedure for the opposite directions. After this process, we obtain a large pseudo-parallel dataset by pairing the best real Nepali sentence with the English input, which we filter out samples whose Levenshtein distances are more than 20% the average sentence length. This means we only accept sentence pairs with very low edit distance.

The sizes of these filtered mined datasets are shown in Table 5. As it can be seen, although the total unfiltered mined datasets are huge at more than 100M pairs, the filtered ones only contains less than 5000 samples for both En-Ne and En-Si. This indicates that because the monolingual corpora between English and low-resource languages are too distant and out-of-domain, there are simply not enough high quality pseudo-parallel data to be mined at all.

Table 5: Back-translated mined pseudo-parallel dataset sizes for English-Nepali and English-Sinhala.

| | En-Ne | Ne-En | En-Si | Si-En |
|---|---|---|---|---|
| **Data information** | | | | |
| Corpus size | En: 100M, Ne: 9.8M | | En: 100M, Si: 8.8M | |
| Unfiltered mined size | ~109.8M | | ~108.8M | |
| Filtered mined size | ~3000 | | ~6000 | |