# OpenReview forum: "Refining Low-Resource Unsupervised Translation by Language Disentanglement of Multilingual Translation Model"
_NeurIPS.cc/2022/Conference — NeurIPS 2022 Accept_

### Official Review · Reviewer_me6V · 2022-06-26

**Rating:** 5
**Confidence:** 4
**Soundness:** 3 good
**Presentation:** 2 fair
**Contribution:** 2 fair

**Summary:**

This paper presents a 4 stage training regime for multilingual unsupervised machine translation. The key idea is to separate the training of a particular component of the  model for each language, and to separate the model training for the into and out of English models.


**Questions:**

Spellcheck please:
4: where theses low-resource languages


**Limitations:**

They discuss the limitations of work that rely on data mining, but do not explicitly address limitations of their own work.

**Strengths And Weaknesses:**

Strengths

They show good improvements over SOTA on 6 low-resource langauge pairs.
They compare with numerous baselines and do ablation studies.

Weaknesses

Complicated strategy involving four stages.
Unsupervised machine translation is an interesting theoretical exercise but in practise you would always use any limited parallel data that you had access to and so it would have been more useful if they had shown their methods benefit low-resource results instead of the unsupervised results.
The authors do not explain their baseline model  LAgSwAV - it is important to understand what we are improving over. They don't make it clear if these baselines (CRISS and LAgSwAV) and their own approach are trained on just the languages shows in their languages families (ie. 3, 2, and 1 langauge pairs) or whether they were trained on the data from all three language groups (ie. 6 language pairs) or on all the data in Commoncrawl 25 langauges. It is therefore hard to interpret the results.
I don't like to have to look at the appendix to see the training data - this is a core part of the paper - but even looking the appendix it was not clear what model was trained on what data.

---

> ### Author Response · Authors · 2022-07-29
> **Our response to the reviewer**
>
> Thanks  for your thoughtful review. Allow us to address your concerns and misunderstanding below.
>
> 1. **About the four-stage strategy**. Our method contains 4 stages, but each stage is easy to understand, simple and requires little modification to the codebase to implement (we released our code with the paper).
> 2. **About using parallel data**. Fully unsupervised MT, where zero parallel data is available, is an established field of research with CRISS(Tran et al., 2020) [CBD (Nguyen et al., 2021)](https://arxiv.org/abs/2006.02163), [mBART (Liu et al., 2020)](https://arxiv.org/pdf/2001.08210.pdf), [MASS (Song et al., 2019)](https://arxiv.org/pdf/1905.02450.pdf), [XLM (Lample et al., 2018)](https://arxiv.org/abs/1901.07291), [NMT&PBSMT (Lample et al., 2018b)](https://arxiv.org/abs/1804.07755), [(Lample et al., 2017)](https://arxiv.org/abs/1711.00043), etc. See [Kyunghyun Cho’s](https://kyunghyuncho.me/) (a program co-chair for NeurIPS-22) keynote at EMNLP-2019 ([slides](https://drive.google.com/file/d/1HGzv6n9hAj-GL63POUZCO6nCrIHF9y35/view) [video](https://www.youtube.com/watch?v=S5mgNwmrutQ)), where he also presents it as an established research problem (slide#30). UMT posits a unique set of challenges to **unsupervised deep learning**. There are [7117 languages](https://www.theintrepidguide.com/how-many-languages-are-there-in-the-world/#:~:text=There%20are%20currently%207%2C117%20known,the%20languages%20of%20the%20world.) in the world, for many of which we can hardly find any parallel data. Unsupervised MT's real-life application expands beyond not only these low-resource languages, but also languages in isolated and remote communities or extinct ancient languages (e.g: ancient Chinese or Egyptian).
> 3. Our work tackles the **research** question of **"Fully unsupervised translation in low-resource languages"**, and not the question of **"How to achieve highest scores possible by gathering as much parallel data as possible"**. Thus, we need to strictly comply with and respect the problem setup to ensure fair comparison with previous established work.
> 4. **About LAgSwAV**: line 27 briefly explains how LAgSwAV works. Basically, it is an unsupervised MT technique that mine pseudo-parallel data from monolingual data by using synthetic cluster assignments [(Caron et al., 2020)](https://proceedings.neurips.cc/paper/2020/file/70feb62b69f16e0238f741fab228fec2-Paper.pdf). We will add more details about LAgSwAV. We appreciate the reviewer's suggestion!
> 5. **Data used**: Section 4.1 explains clearly that we use the Common Crawl (CC25) dataset of 25 languages, which was the exact same dataset used to train CRISS and LAgSwAV. Since we finetune our model from CRISS, our model was already pre-trained on the entire data from all 25 languages. Our method (the 4 stages) only selects a portion of the same dataset of relevant languages to finetune the model and boost the performance. Thus, **essentially, no new or different data were used to train the model as a whole, compared to the baselines.**
> 6. **Limitation**: We make it clear in different occasions in the paper (line 52,98,99,100,478) that our limitation is that our method prioritizes to maximize the individual performance of each low-resource task by making the trade-off that it can no longer translate multiple languages in a one-for-all manner.

---

### Official Review · Reviewer_5Niy · 2022-07-06

**Rating:** 6
**Confidence:** 3
**Soundness:** 3 good
**Presentation:** 3 good
**Contribution:** 3 good

**Summary:**

The paper proposes a 4-stage refinement of a multi-lingual unsupervised NMT model into direct NMT models for low-resource languages, in a restricted setting when there is zero parallel data. While the procedure and its implementation is rather complicated, it delivers large BLEU gains.


**Questions:**

- 327: in what way LASER uses the encoded sentences from CRISS?
- If possible, please consider replacing "disentanglement" with "refinement" or other word for the final version, since in ML disentanglement has a specific meaning of learning causal factors in latent spaces of generative models.


**Limitations:**

Focus on unrealistic setup with exactly zero parallel data available for all languages

**Strengths And Weaknesses:**

Strengths:
- High improvements in the zero-parallel-data setting
- Solid comparisons to SOTA and ablation experiments
- Very well written paper

Weaknesses:
- While the authors make sure to carefully delineate their work from UMT that uses some quasi-parallel data, this reduces the value of the work by constraining it into an unrealistic setup -- in practice, even if public corpora like CC have too little parallel data to be mined, there are always e.g. religious texts that have more than 6000 sentences. Along the same lines, a more relevant to practice comparison would be to compare systems with existing parallel texts added.
- The paper may be a weak fit for NeurIPS, as it focuses on a niche MT topic

---

> ### Author Response · Authors · 2022-07-29
> **Our response to the reviewer**
>
> Thanks  for your thoughtful review. Allow us to address your concerns and misunderstanding below.
>
> 1. Fully unsupervised MT (UMT), where zero parallel data is available, is an established field of research with methods such as CRISS (Tran et al., 2020) [CBD (Nguyen et al., 2021)](https://arxiv.org/abs/2006.02163), [mBART (Liu et al., 2020)](https://arxiv.org/pdf/2001.08210.pdf), [MASS (Song et al., 2019)](https://arxiv.org/pdf/1905.02450.pdf), [XLM (Lample et al., 2018)](https://arxiv.org/abs/1901.07291), [NMT&PBSMT (Lample et al., 2018b)](https://arxiv.org/abs/1804.07755), [(Lample et al., 2017)](https://arxiv.org/abs/1711.00043). See [Kyunghyun Cho’s](https://kyunghyuncho.me/) (a program co-chair for NeurIPS-22) keynote at EMNLP-2019 ([slides](https://drive.google.com/file/d/1HGzv6n9hAj-GL63POUZCO6nCrIHF9y35/view) [video](https://www.youtube.com/watch?v=S5mgNwmrutQ)), where he also presents it as an established research problem (slide#30). UMT posits a unique set of challenges to **unsupervised deep learning**. There are lots of research interests in this field! There are [7117 languages](https://www.theintrepidguide.com/how-many-languages-are-there-in-the-world/#:~:text=There%20are%20currently%207%2C117%20known,the%20languages%20of%20the%20world.) in the world, and there are simply not enough religious texts to cover most of them. UMT's real-life application expands beyond not only these low-resource languages that are spoken in countries with little connection with the English language, but also languages in isolated and remote communities or extinct ancient languages (e.g: ancient Chinese or Egyptian).
> 2. The research area of UMT started with high-resource language, like En-Fr in (Lample et al., 2017), (Lample et al., 2018), (Lample et al., 2018), and gradually moved on to more challenging and practical low-resource languages (CRISS, LAgSwAV, mBART). Our work continues to contribute to such a foundation.
>
> 3. We respectfully disagree with the argument that this topic is a weak fit for NeurIPS! As mentioned in (1), this is an important machine learning problem (particularly **unsupervised deep learning**) that fits very well in NeurIPS and alikes. In fact, many of previous papers that we cite and compare with were published here or similar venues: (Lample et al., 2017) in ICLR 2018; MASS (Song et al., 2019) in ICML 2019, CBD (Nguyen et al., 2021) in ICML 2021, CRISS (Tran et al., 2020) in NeurIPS 2020, LAgSwAV (Nguyen et al., 2022) in ICLR 2022.
>
> Questions:
>
> 1. LASER uses nearest-neighbor clustering from the embeddings of sentences generated by CRISS's encoder (by average pooling hidden vectors of all tokens). Nearest-neighbor clustering will determine the closest cross-lingual pair of data from the embedding space.
>
> 2. Thanks for the suggestion; we will use "separation" instead of "disentanglement" in the final version.

---

### Official Review · Reviewer_hAiK · 2022-07-10

**Rating:** 7
**Confidence:** 5
**Soundness:** 2 fair
**Presentation:** 4 excellent
**Contribution:** 3 good

**Summary:**

This paper presents a method to improve unsupervised neural machine . They propose to disentangle languages learned by multilangual language modes (eg. mBART). It is done through 4 different steps all explained in dedicated subsections. The method is evaluated on several language pairs from different language families, but always involving English. The paper claims to achieve the state-of-the-art on some of them.

The proposed method is very interesting, intuitive, innovative, and well-motivated. I especially find the step one very clever, though a bit complicated, giving each language pair its own set of layers, fine-tuned on individual GPUs. It is well engineered to remain efficient, and the authors even discussed the case where only 1 GPU is available. It is clear why it should work to improve machine translation for these language pairs.

However, the evaluation is extremely sloppy... Most of the comaprisons with previous work is done without checking whether all the BLEU scores compared are truly comparable. Actually, they are not, and this makes half of the paper incorrect.
I recommend to reject the paper unless the authors can convince us that all (not juste one or two) the systems compared used: the same preprocessed data, the same reference translations, the exact same tokenizer, and the same multi-bleu.perl. This is still surprising to see this kind of evaluation in 2022, especially for authors using the script multi-bleu.perl that clearly warns when executed to not publish multi-bleu scores. Why not using sacreBLEU? or, even better, COMET?
I recommend the authors to read Marie et al. (2021)'s meta-evaluation of MT to fully understand what make their evaluation not credible.

The proposed method is very interesting an I enjoyed reading the paper, but please correct your evaluation.

**Questions:**

1. it would be nice to have one figure presenting the 4 first steps and their connection to help the reader understands how it works.
2. Section 4.1: where did you get the common-crawl data exactly and how did you extract it? Is this the same common-crawl data used by all the baselines systems? If not, how can you be sure, following the scientific methodology, that the improvement comes from your method and not from the data?
3. How does the proposed method compares to 'Multilingual Unsupervised Neural Machine Translation with Denoising Adapters'? It should be your baseline.

To be clear, if you correct the evaluation and add Bérard et al. (2021) as baseline (or at least explain why it is not a relevant baseline), I will change my score to a 7 or 8. For now, I can only recommend to reject the paper.

**Limitations:**

- yes the authors addresses the limitations of their work in a dedicated subsections.

**Strengths And Weaknesses:**

Strength:

- well-written
- perfectly motivated: I fully understand why the authors worked on this method
- well-engineered: the proposed architecture seems to remain efficent with some clever engineering tricks
- probably works to improve machine translation quality for the targeted language pairs

Weaknesses:

- the evaluation is incorrect. BLEU scores from different papers using different data, tokenization, etc are compared.
- the method looks complicated (but is not complicated), mostly due to its decomposition into 4 different steps that are not that easy to connect. I think it should and can be reformulated as a one single step method.
- a very close work is not cited: Multilingual Unsupervised Neural Machine Translation with Denoising Adapters (Bérard et al, 2021) and should probably be the baseline of this paper. It aims at achieving the same goal but through a different method. Bérard et al, (2021) looks much simpler.

---

> ### Author Response · Authors · 2022-07-29
> **We address the reviewer's concerns - Regarding the BLEU scores**
>
> Thanks  for your thoughtful review. Allow us to address your concerns and misunderstanding below. Our response consists of 3 sections: _BLEU evaluation_, _(Bérard et al, 2021) comparison_ and _other questions_. We hope that it adequately addresses your concerns and we urge you to reconsider your decision and scores. We will be also happy to answer if you have further questions/concerns.
>
> ## Regarding BLEU evaluation
>
> **We respectfully confirm with ABSOLUTE CERTAINTY that the ALL BLEU evaluation is correct and comparable:**
>
> **How we make sure it is correct and comparable:**
>
> 1. Even before we started working on this paper, **we did communicate with the authors of CRISS (Tran et al., 2020) and LAgSwAV (Nguyen et al., 2022)** to cross-check and obtain their test sets with reference translations, preprocessing scripts, tokenization process, and the _multi-bleu.perl_
> 2. We have been using their evaluation pipeline from the start and we did not change anything regarding this.
> 3. To confirm consistency further, we obtained their pretrained models and conducted BLEU evaluaton on the test sets and confirmed that it is consistent and reproducible with what they reported.
> 4. **Our project was developed from their [codebase](https://github.com/facebookresearch/fairseq/tree/main/examples/criss), using their evaluation scripts and test sets.**
> 5. Their evaluation pipeline (test sets, reference translations, preprocess scripts and tokenziation) is in fact inherited from previous unsupervised MT work in the literature: [CBD (Nguyen et al., 2021)](https://arxiv.org/abs/2006.02163), [mBART (Liu et al., 2020)](https://arxiv.org/pdf/2001.08210.pdf), [MASS (Song et al., 2019)](https://arxiv.org/pdf/1905.02450.pdf), [XLM (Lample et al., 2018)](https://arxiv.org/abs/1901.07291), [NMT&PBSMT (Lample et al., 2018b)](https://arxiv.org/abs/1804.07755), and [(Lample et al., 2017)](https://arxiv.org/abs/1711.00043).
>
> **Why we HAD TO use multi-bleu.perl:**
>
> We thank the reviewer and appreciate your suggestions in bringing up the issue with multi-bleu.perl! We sincerely understand the reviewer's frustration. We do not like it either!
> But as the CRISS's authors mentioned in their paper: "To be consistent with previous literature, we used multi-bleu.perl for evaluation.". Despite the push towards the use of SacreBLEU, the literature of unsupervised machine translation (e.g: CRISS, LAgSwAV, CBD (Nguyen et al., 2021), [mBART (Liu et al., 2020)](https://arxiv.org/pdf/2001.08210.pdf), [MASS (Song et al., 2019)](https://arxiv.org/pdf/1905.02450.pdf), [XLM (Lample et al., 2018)](https://arxiv.org/abs/1901.07291), [NMT&PBSMT (Lample et al., 2018b)](https://arxiv.org/abs/1804.07755), and [(Lample et al., 2017)](https://arxiv.org/abs/1711.00043)...) has been sticking with multi-bleu.perl since 2018 (the year SacreBLEU is published). Subsequent work have been using multi-bleu to ensure consistency with previous work. And of course, they still use it in 2022 [(Nguyen et al., 2022)](https://openreview.net/pdf?id=pN1JOdrSY9).
>
> We hope that the reviewer will consider that it is certainly **not our fault** for following the literature of the field, and we believe punishing our paper for this would be unfair.  If we had reported SacreBLEU, the results for CRISS and LAgSwAV would be different from what was reported, and we fear that there would be complaints and doubts about such inconsistency. We thus urge the reviewer to have an empathetic view of the field our work is in, and how it fits and compares to peers in the field of unsupervised machine translation.
>
> **Despite that, as you suggested, we report SacreBLEU scores in the tables below, and have also added these numbers to our revised paper**
>
> We thank the reviewer and appreciate your suggestions in bringing up the issue with multi-bleu.perl!
>
> Indic languages
>
> |                 | En-Ne | Ne-En | En-Si | Si-En | En-Hi | Hi-En | En-Gu  | Gu-En |
> | -- | --- | --- | --- | --- | --- | --- | --- | --- |
> | LAgSwAV         | 5.3   | 12.8  | 5.4   | 9.4   | - | -| - | -  |
> | CRISS           | 5.5   | 14.5  | 6.0   | 14.5  | 19.4  | 23.6  | 14.2   | 23.7 |
> | Ours            | 9.0   | 18.2  | 9.5   | 15.3  | 20.8  | 23.8  | 17.5   | 29.5 |
> |LAgSwAV-SacreBLEU| 5.4   | 12.5  | 5.3   | 9.1   | - | -| - | - |
> | CRISS-SacreBLEU | 5.6   | 14.4  | 6.0   | 14.3  | 19.5  | 23.1  | 14.2   | 23.4 |
> | Ours-SacreBLEU  | 9.1   | 18.1  | 9.4   | 15.1  | 20.9  | 23.4  | 17.5   | 29.1 |
>
>
>
> Uralic
>
> |         | En-Fi | Fi-En | En-Lv | Lv-En | En-Et | Et-En |
> | -- | -- | - | -- | -- | -- | -- |
> | CRISS | 20.2 | 26.7 | 14.4  | 19.2 | 16.8 | 25.0 |
> | Ours | 22.9 | 28.2 | 18.5 | 19.3 | 21.0 | 25.7 |
> | CRISS-SacreBLEU | 20.1 | 26.2 | 14.3  | 18.6 | 16.7 | 24.6 |
> | Ours-SacreBLEU | 22.8 | 27.8 | 18.2 | 18.9 | 20.9 | 25.4 |
>
> Turkic
>
> |   | En-Kk  | Kk-En |
> | -- | -- | -- |
> | CRISS | 6.7 | 14.5 |
> | Ours | 10.0 | 20.0 |
> | CRISS-SacreBLEU | 6.7 | 14.1 |
> | Ours-SacreBLEU | 10.0 | 19.7 |
>
> # Response continues below !

---

> > ### Author Response · Authors · 2022-07-29
> > **Continued response - Regarding why (Bérard et al, 2021) is not a comparable baseline and other questions**
> >
> > ## Regarding why (Bérard et al, 2021) is not a comparable baseline
> >
> > In short, [(Bérard et al, 2021)](https://arxiv.org/pdf/2110.10472.pdf) cannot be fairly and correctly compared with our work (as well as prior UMT papers such as CRISS, LAgSwAV, MASS and XLM) because it **uses auxiliary parallel (supervised) data** to train their model! While our fully unsupervised setup does not use any supervision of any kind in any stages of the training process (including pre-training of CRISS).
> >
> > We are fully aware of (Bérard et al, 2021). This work belongs to a paradigm of methods that improve MT of a target language pair (e.g., En-Fr) by using auxiliary supervised parallel data from another related pair (e.g., En-De). Their definition of “unsupervised translation” is not consistent with the setup proposed in the mainstream UMT work  [(Lample et al., 2017)](https://arxiv.org/abs/1711.00043), [(Lample et al., 2018b)](https://arxiv.org/abs/1804.07755), [(Lample et al., 2018)](https://arxiv.org/abs/1901.07291), [(Song et al., 2019)](https://arxiv.org/pdf/1905.02450.pdf), [(Nguyen et al., 2021)](https://arxiv.org/abs/2006.02163), etc. To our knowledge, this is first proposed and popularized by [(Garcia et al., 2020)](https://aclanthology.org/2020.findings-emnlp.283/) and [(Garcia et al., 2021)](https://aclanthology.org/2021.naacl-main.89/). By using supervision from a related language pair, this line of work is **not fully** unsupervised MT, rather they can better be framed as **zero-shot MT**, which means any kind of supervision is allowed as long as there is no supervision in the target language pair.
> >
> > As [(Guzman et al., 2019)](https://aclanthology.org/D19-1632.pdf) pointed out, using related parallel data to aid a target language pair is poised to pump the BLEU significantly regardless of whether the methodology is better or not, thus giving this a significant and unfair advantage.
> >
> > For the FLoRes Nepali-En and Sinhala-En tasks, Bérard et al, (2021) trained their model with supervised parallel data in Hindi-En, without considering that Hindi and Nepali share a significant amount of lexicons and grammars (in fact, **Hindi speakers can understand certain Nepali texts**).
> >
> > Note that the [FLoRes (Guzman et al., 2019)](https://aclanthology.org/D19-1632.pdf) also applies Hi-En parallel data on unsupervised Ne-En and achieves 18.8 BLEU, outperforming (Bérard et al, 2021) itself with 18.1 BLEU. This is despite the fact that Bérard et al, (2021) claims to have a superior method.
> >
> > We dedicated a paragraph (line 112-119) to discuss this issue and differentiate our fully unsupervised field of work and theirs. We did cite (Garcia et al., 2020,2021). We will cite (Bérard et al, 2021) too, as suggested.
> >
> > **But as suggested, we compare our method with (Bérard et al, 2021) as below**. Note that (Bérard et al, 2021) unfairly uses supervised parallel data on 20 language pairs (en with ar,he,ru,ko,it,ja,zh,fr,pt,tr,ro,pl,vi,de,fa,cs,th,my,hi):
> >
> > |         | Ne-En | Si-En | En-Fi | Fi-En | En-Et | Et-En |  En-Kk  | Kk-En |
> > | --- | --- | --- | --- | --- | --- | --- | --- | --- |
> > |(Bérard et al, 2021) | 18.1 | 11.4 | 11.7 | 25.7 | 17.1 | 29.3 | 4.9 | 16.4 |
> > | Ours-SacreBLEU | 18.1  | 15.1 | 22.8 | 27.8 | 20.9 | 25.4 |10.0 | 19.7 |
> >
> > #### Other questions and concerns:
> >
> > 1. Figure of steps: Figure 2 presents the 4 stages of the method and how they connect. We will annotate this diagram with more details for better understanding (see the revised version).
> >
> > 2. As explained above, we make sure we use the same dataset, test sets, codebase and evaluation scripts as CRISS. [The training set](https://data.statmt.org/cc-100/) [(Wenzek et al., 2020)](https://aclanthology.org/2020.lrec-1.494.pdf) is the same, and has been used not by not only CRISS, but mBART and [XLM-R](https://aclanthology.org/2020.acl-main.747.pdf). As our method is finetuned from CRISS, we actually use a small fraction of the same available data.

---

> > > ### Comment · Reviewer_hAiK · 2022-08-03
> > > **About the unsupervised setting**
> > >
> > > This part of your answer is outstanding.
> > >
> > > I fully agree that Berard et al work is not fully unsupervised. Indeed, I'm influenced by previous work, notably by Garcia et al., who claim to be unsupervised even when using 'auxiliary parallel data'. Not using it as a baseline is definitely acceptable. I accept it. Please be sure to clearly point out that in contrast with previous work, your work is really fully unsupervised. Otherwise you expose yourself to receiving comments from researchers that do not see Garcia et al work, and the following papers improving on it, as zero-shot. Some do see this line of work using auxiliary parallel data as 'unsupervised'.

---

> > ### Comment · Reviewer_hAiK · 2022-08-03
> > **Answer about BLEU**
> >
> >
> > This is a fantastic answer!
> >
> > Thank you for the time you took to construct your argumentation. I am convinced that the scores are probably comparable given all the supplementary details you mentioned. I suggest you add in the paper that you had personal communications with the author of previous work to confirm the evaluation settings, otherwise I don't think it would have been possible to confirm it since base codes are usually updated, and usually do git clone of the preprocessing tools (tokenizer) that may also be updated and thus won't remain identical to the original version released with the paper. I'm sure you did your best to confirm that everything is the same as in the previous work you are comparing with.

---

> > > ### Author Response · Authors · 2022-08-04
> > > **Our response to the reviewer**
> > >
> > > We sincerely thank you for reconsidering your decision and greatly appreciate your responses. We will follow your suggestions to add these details about how evaluation was done in the paper. You are completely right in that sometimes codebases and dependencies cloned from git are changed after previous work was published, and the previous authors may not even notice such changes.
> > >
> > > In our case, luckily they provided the pre-trained models, so we are able to perform an additional check to confirm the consistency of the evaluation pipeline by reproducing their reported results with the pre-trained models.

---

### Official Review · Reviewer_yLsx · 2022-07-13

**Rating:** 6
**Confidence:** 4
**Soundness:** 2 fair
**Presentation:** 3 good
**Contribution:** 3 good

**Summary:**

This work proposes a low resource machine translation in unsupervised setting in which languages in the target side is explicitly disentangled from a multilingual model. Basic idea is to employ target language specific feed-forward layers in the decoder side in order to isolate the language specific representation. The model is trained in multiple stages:
1. Starting from a multilingual model trained in an unsupervised way, e.g., CRISIS, the model is trained jointly by augmenting training data using back-translation.
2. The model is split into two, English-to-X and X-to-English, and they are trained individually starting from the parameters learned in the first step. This step is iterated twice.
3. The model for English-to-X is further split into target language specific models so that the representation is further isolated in the target side.

This procedure results in a single X-to-English model and multiple English-to-X models. Experimental results on standard unsupervised machine translation benchmark shows significant gains when compared with multilingual machine translation baselines.

**Questions:**

* Table 2 is strange in that no training happens in stage 4 for X-to-English direction, but BLEU scores are slightly different. I feel something wrong in the description in this submission or a bug in experiments. Or, there's somethings I've completely missed.

**Limitations:**

* The proposed method does not train a multilingual model in English-to-X directions, and, thus, spuriously many parameters must be learned for each direction.

**Strengths And Weaknesses:**

Strengths
* The idea itself is not novel in that there exists prior work on isolating target language specific representation by feed-forward network in the decoder side. However, its application for unsupervised setting is a new contribution.
* The training algorithm is carefully designed to empirically avoid poor local optima by splitting into multiple stages.
* This work also proposes efficient training by exploiting multiple GPUs so that the parameters for language specific feed-forward layers do not have to be shared among multiple devices.

Weaknesses
* ~~The efficient training strategy is applied only in the first stage of training, i.e., training a single multilingual model with target language specific parameters.~~ Thanks for the clarification, but it is applicable when multiple languages are involved, and loosing its merit in the later steps when tuning for bilingual model.
* It is not clear whether the comparison is meaningful in that the proposed model is basically learning language pair specific model for English-to-X direction. Other models, e.g., CRISSS, are basically multilingual models so that a single model is employed to translate multiple directions. This work should be compared with other bilingual baselines that rely on unsupervised method. Or, it would be meaningful to iterate the first stage multiple times to see if there exists any gains without explicitly separating parameters.

---

> ### Author Response · Authors · 2022-07-29
> **Response to reviewer's concerns**
>
> Thanks for your thoughtful review. Let us clarify your concerns:
>
> 1. **Efficient training**: The efficient training strategy with language-specific FFNs is applied not just in the 1st, but also in the 2nd and 3rd stages, though not in the last one. In the 2nd and 3rd stages, the model is split into 2 models (to-English and from-English)
> 2. **Comparison**: To our knowledge, CRISS is the best-performing fully unsupervised model in the market to compare with for these low-resource languages. As far as we know, there is no bilingual unsupervised baseline that can perform anywhere close to CRISS in these languages (multilingual training helps bootstrap low-resource performance). Bilingual unsupervised baselines such as XLM (Lample et al. 2018) and MASS (Song et al., 2019) give near 0 BLEU for these MT tasks due to the low-resource nature of these languages, as shown in (Guzman et al., 2019). **In other words, our method is compared with and outperforms, not only the most comparable but also the state-of-the-art unsupervised model that is available in the market.** We would be happy to consider any particular bilingual model you have in mind.
> 3. **Stage-1 Iteration**: Regarding iterating the first stage multiple times, we actually did that experiment (iterate 5 times)! The results were not better than the "+Stage 1" of Table 2. We observe significant overfitting and performance drop instead. This is one of the reasons why we develop the other 3 stages, which we prove to be necessary to improve the performance significantly. **We will mention this in the paper.**
> 4. **Table 2** is meant to study the impact of different stages on the performance. Thus, as we made it clear in  Section 4.2 (line 262-267): **we still applied _Stage-4 training to X-to-English anyway_ to study its impact, despite the fact that Stage-4 on X-to-English is not applied in our main system (like in Table 1)**. As a result, the numbers for X-to-English for "+Stage 4" in Table 2 are slightly different from Table 1 and "+Stage 3" in Table 2.
>
> We hope our responses clarify your concerns and misunderstanding. We sincerely hope you reconsider your decision.

---

### Author Response · Authors · 2022-07-30
**Summary of authors' responses and submission of updated rebuttal paper.**

Dear Reviewers and ACs,

We sincerely appreciate the reviewers' effort in reviewing our paper and the AC for reviewing and organizing discussions!

We have submitted specific responses that address each reviewer's concerns and questions. In relation to that, we have submitted an updated version (rebuttal version), which contains changes that address the reviewers' concerns and incorporate the valuable feedback that the reviewers provided.

The changes we made include:

1. SacreBLEU numbers in Table 1 as subscripts to the main numbers to provide more details.

2. New annotations and explanation in Figure 2 to provide a better description of the method.

3. More explanation to the baseline (LAgSwAV).

As we continue improving the paper, this will not be the final version and we hope to receive more feedback in the discussion.

In addition, we would like to highlight a few key points that arise in the reviews:

1. **BLEU evaluation:** multi-bleu.perl was used to evaluate and compare our method with the baselines (like CRISS or LAgSwAV) exclusively because these baselines also used multi-bleu.perl to ensure consistency with the literature of the field [MASS (Song et al., 2019)](https://arxiv.org/pdf/1905.02450.pdf), [XLM (Lample et al., 2018)](https://arxiv.org/abs/1901.07291), [NMT&PBSMT (Lample et al., 2018b)](https://arxiv.org/abs/1804.07755), and [(Lample et al., 2017)](https://arxiv.org/abs/1711.00043). To make sure the correct use of multi-bleu.perl, we took significantly rigorous steps (see our response to reviewer hAiK) to ensure all steps of the evaluation process (from datasets, tokenization, models...) are consistent with the baselines. We believe these steps, along with reporting additional SacreBLEU numbers, are sufficient to not only respect (and be consistent with) the work before us, but also provide more context to other researchers.

2. The research problem we consider is **fully unsupervised machine translation**, where the most important requirement is **no** parallel data or supervision of any kind is used anywhere from scratch to end model. This field is established through various progressive work by [(Tran et al., 2020)](https://arxiv.org/pdf/2006.09526.pdf), [CBD (Nguyen et al., 2021)](https://arxiv.org/abs/2006.02163), [(Liu et al., 2020)](https://arxiv.org/pdf/2001.08210.pdf), [(Nguyen et al., 2021)](https://arxiv.org/abs/2006.02163), [(Liu et al., 2020)](https://arxiv.org/pdf/2001.08210.pdf), [(Song et al., 2019)](https://arxiv.org/pdf/1905.02450.pdf), [(Lample et al., 2018)](https://arxiv.org/abs/1901.07291), [(Lample et al., 2018b)](https://arxiv.org/abs/1804.07755), and [(Lample et al., 2017)](https://arxiv.org/abs/1711.00043). Meanwhile, there is another line of work that utilizes parallel and supervised auxiliary data from a related language pair to support the target pair in a zero-shot fashion [(Bérard et al, 2021)](https://arxiv.org/pdf/2110.10472.pdf), [(Garcia et al., 2020)](https://aclanthology.org/2020.findings-emnlp.283/), [(Garcia et al., 2021)](https://aclanthology.org/2021.naacl-main.89/). Though useful and appreciative in its own right, this setup is not within the scope of the problem we are pursuing, as introducing related parallel data induces an unfair advantage against our setup [(Guzman et al., 2019)](https://aclanthology.org/D19-1632.pdf) and jeopardises our ability to recognize the source of improvement. Therefore, we believe using parallel data, even with the smallest amount, or comparing with the work mentioned above, are not within the scope of our paper and shall be tackled separately.

We hope to get valuable comments from the reviewers and we are ready and glad to engage in active discussions.

---

### Author Response · Authors · 2022-08-07
**We hope to have further discussion**

Dear reviewers,

We greatly appreciate the reviewers’ effort and time in reviewing our paper. We have submitted our responses to each reviewer and uploaded a rebuttal revision of the paper with details of the changes. While we are greatly grateful that reviewer *hAiK* has replied and discussed our responses, we hope to also have further discussions and gather feedback from the other reviewers (yLsx, 5Niy and me6V) as well.

As the deadline for the discussion period is ending soon, and we will not be able to discuss with the reviewers further, we sincerely hope to receive a response or acknowledgement from the other reviewers.

Thank you.

---

### Meta-Review · Area_Chair_XhvG · 2022-08-28

**Recommendation:** Accept
**Confidence:** Certain

**Metareview:**

This paper presents a four-stage process for training completely unsupervised machine translation models. The results are fairly strong. After some discussion all reviewers are convinced that the evaluation is sound.
The reviewers are somewhat split about the novelty, some say:
+ "The proposed method is very interesting, intuitive, innovative, and well-motivated."
Others say:
- "The idea itself is not novel in that there exists prior work on isolating target language specific representation by feed-forward network in the decoder side."
Some reviewers find the approach with its four steps cumbersome, while others don't have an issue with this.
On balance, just above the decision boundary.

**Award:**

No

---

### Decision · Program_Chairs · 2022-09-14

Accept